



# Tree island area in oil palm agroforests directly and indirectly drives evaporative fraction

Thorge Wintz[1], Alexander Röll[2], Gustavo Brant Paterno[3], Florian Ellsäßer[4], Delphine Clara Zemp[5],
Hendrayanto[6], Bambang Irawan[7], Alexander Knohl[8,9], Holger Kreft[3,9], Dirk Hölscher[1,9]

[1] Tropical Silviculture and Forest Ecology, University of Göttingen, Büsgenweg 1, 37077 Göttingen, Germany
[2] Horticultural Sciences, Institute for Crop Science and Resource Conservation, University of Bonn, Auf dem Hügel 6, 53121 Bonn, Germany
[3] Biodiversity, Macroecology & Biogeography, University of Göttingen, Büsgenweg 1, 37077 Göttingen, Germany
[4] Department of Natural Resources, ITC, University of Twente, 7522 NH Enschede, Netherlands
[5] Conservation Biology, Institute of Biology, Faculty of Sciences, University of Neuchâtel, Neuchâtel, Switzerland
[6] Faculty of Forestry, Bogor Agricultural University, Campus Darmaga Bogor, Jawa Barat 16680, Indonesia
[7] Forestry Department, Faculty of Agriculture, University of Jambi, Jambi, 36122, Indonesia
[8] Bioclimatology, University of Göttingen, Büsgenweg 2, 37077 Göttingen, Germany
[9] Centre of Biodiversity and Sustainable Land Use, University of Göttingen, Büsgenweg 1, 37077 Göttingen, Germany

*Correspondence to*: Thorge Wintz (twintz@uni-goettingen.de)





Graphical abstract:

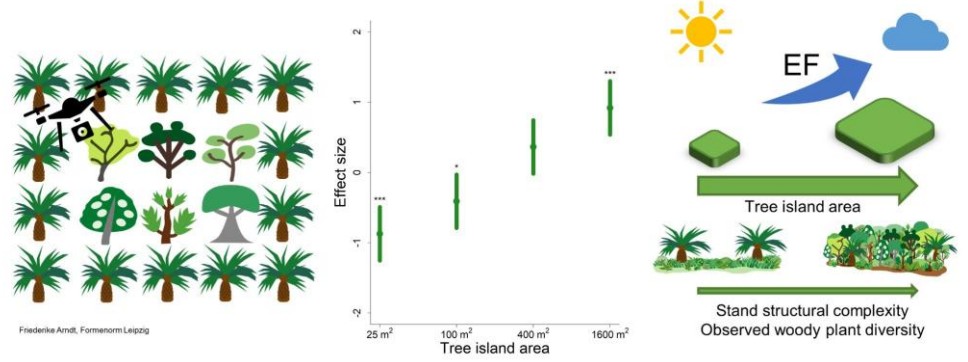

35





**Abstract.** Evapotranspiration (ET) – the combined water flux from soil and vegetation to the atmosphere – is a key component of water cycling and climate regulation, and strongly affected by land-use changes. The evaporative fraction (EF), representing the proportion of available energy allocated to ET, is often preferred over ET as a target variable in studies involving repeated measurements under varying weather conditions. In Sumatra's lowlands in Indonesia, (evapo)transpiration of dominant land-use types including oil palm monocultures is well studied; however, there is a lack of studies assessing ET (or EF) across diverse mosaic landscapes and types of land-use such as oil palm agroforests. Across 52 experimental plots – forest restoration patches known as "tree islands" – in an oil palm landscape (EFForTS-BEE), we tested whether the experimental treatments 'planted tree diversity' and 'tree island area' influence ET and EF as derived from UAV (uncrewed aerial vehicle)-based thermography and subsequent energy balance modeling. A random partition linear model showed that planted tree diversity (1, 2, 3, or 6 species) did not affect plot-level ET or EF, whereas tree island area (25, 100, 400, or 1600 m²) had a positive effect, with EF increasing by 17% from the smallest to the largest tree islands. A structural equation model revealed that the effect of tree island area on EF was mediated by both direct and indirect pathways. Specifically, a strong direct effect of island area on EF (Std.Beta = 0.44, $p < 0.001$) was complemented by an indirect pathway through increased observed woody plant diversity and stand structural complexity. Stand structural complexity had a positive effect on EF (Std.Beta = 0.20, $p < 0.05$), while neither the vegetation index GNDVI nor tree height variability had significant effects. The observed tree-island-area effect can be explained by a decrease of EF along an edge gradient detected inside the larger tree islands. Our findings suggest that larger tree islands enhance ET and EF through structural and biodiversity-related mechanisms. This underscores the importance of tree islands in human-modified landscapes, not only as biodiversity refugia but also as functional elements that support climate regulation.

Key words: patchy landscapes, edge effect, evapotranspiration, evaporative cooling, land surface temperatures, UAV, TreeDivNet



## 1 Introduction

The conversion of tropical rainforests is ongoing and impairs biodiversity, ecosystem functioning and services (Edwards et al., 2013; Grass et al., 2020; Potapov et al., 2024). Commodity-driven deforestation is a major driver and often leads to landscapes dominated by large-scale monocultural plantations. In recent decades, oil palm cultivation has expanded, particularly in the humid tropics, with Indonesia being the world's largest producer of palm oil (FAO, 2023). Oil palm cultivation contributes to food security, improves the livelihoods of local people, and enables macro-economic development (Kubitza et al., 2018; Qaim et al., 2020). However, it has also contributed to a massive decline in biodiversity and has compromised the carbon and hydrological cycles (Kotowska et al., 2015; Merten et al., 2016; Guillaume et al., 2018; Grass et al., 2020), as well as the microclimatic regulation of air and land-surface temperatures (Meijide et al., 2018; Pallavi et al., 2024). Consequently, there is an intensive search for more sustainable modes of oil palm cultivation, which may include reduced application of agrochemicals (Iddris et al., 2023) or the integration of trees into the cultivation system (Teuscher et al., 2016; Zemp et al., 2023; Wenzel et al., 2024).

Tree islands, i.e., local tree plantings of native species within the oil palm-dominated landscape, are being studied within the Biodiversity Enrichment Experiment (EFForTS-BEE) in lowland Sumatra, Indonesia. The experiment consists of 52 experimental tree islands of varying tree diversity and area, distributed across an oil palm-dominated landscape. The experimental enrichment of tree diversity was motivated by the generally positive relationship between biodiversity and ecosystem functioning (Grossman et al., 2018). Tree diversity usually enhances productivity, ecosystem carbon storage, transpiration and decomposition rates (Paquette and Messier, 2011; Kunert et al., 2012; Zemp et al., 2023; Zheng et al., 2024). The treatment island area was motivated by the theory of island biogeography, which states that larger islands harbor more species than smaller islands, driven by species colonization and extinction (MacArthur and Wilson, 1963; Gooriah et al., 2020). This theory also plays a major role in nature conservation (Givnish, 2024), for example regarding the size of protected areas or fragments in patchy landscapes (Arroyo-Rodríguez et al., 2020).

Establishing the tree islands in the experimental EFForTS-BEE landscape had pronounced positive effects on biodiversity, for example by increasing tree diversity by +4.7 compared to oil palm monoculture, and on key ecosystem functions such as water infiltration, which was achieved without reducing landscape-level oil palm yields (Zemp et al., 2023). Both treatments – planted tree diversity and in particular island area – enhanced the taxonomic, phylogenetic and functional diversity of recruiting native woody species (Paterno et al., 2024). While EFForTS-BEE revealed considerable potential to alleviate biodiversity decline and impaired ecosystem functions by integrating tree islands into oil palm landscapes, a comprehensive assessment regarding evapotranspiration (ET) is still missing.



ET is an ecosystem function that contributes to climate regulation and water cycling. The prominent role of the tropical
rainforest in global evaporative cooling was highlighted by Lawrence et al. (2022), but this function is threatened by forest
conversion to other land-use types. However, ET from oil palm, particularly from mature and intensively managed plantations,
has been reported to match or even exceed that of tropical forests (Sabajo et al., 2017; Manoli et al., 2018; Röll et al., 2019;
Gómez et al., 2023) – a pattern similarly observed in tree plantations and early successional tree stands, which can also exhibit
high ET rates (Dierick et al., 2010). Landscape fragmentation and increasing edge density can influence the species
composition of forest remnants by favoring certain functional traits while disfavoring others (Pinho et al., 2024). These trait
shifts can influence ecosystem functioning, potentially affecting processes like water use and energy partitioning. Islands
intrinsically comprise edges, and edges can have enhanced transpiration rates compared to the interior (Giambelluca et al.,
2003; Kunert et al., 2015). This effect might play a role when comparing the average ET of tree islands which differ in edge
length.


Assessments of ET from forests or other land-use types in human-modified landscapes are often based on eddy covariance or
sap flux measurements (Niu et al., 2015; Meijide et al., 2017). Their strengths include the provisioning of high-resolution time
series, potentially also over long periods. Weaknesses include the restriction to a relatively low number of localities and large
measurement footprints mixing multiple landscape types or entities such as trees. It is therefore difficult to assess patchy
landscapes such as EFForTS-BEE, with its 52 tree islands and more than 10 000 trees, using such methods. At such large
scales, remote sensing approaches may be more feasible (Senf, 2022), for example using uncrewed aerial vehicles (UAV) as
a monitoring platform. ET derived from UAV-thermography and subsequent energy balance modeling has previously been
applied in horticulture (Park et al., 2021; Riveros-Burgos et al., 2021) and forests (Bulusu et al., 2023; Cortés-Molino et al.,
2024). A well-established approach uses maps of land surface temperatures and solar radiation data to predict ET with the
DATTUTDUT (Deriving Atmosphere Turbulent Transport Useful To Dummies Using Temperature) energy balance model
(Timmermans et al., 2015). This approach has been implemented in a freely available QGIS Plugin, QWaterModel (Ellsäßer
et al., 2020b). In a mature oil palm plantation of lowland Sumatra, it was tested against eddy covariance measurements
(Ellsäßer et al., 2021). The derived estimates of latent heat flux agreed very well between the two methods across variable
weather conditions and times of day, and further analyzes even suggested statistical interchangeability (Ellsäßer et al., 2021).
The same UAV-based approach was also successful in predicting stomatal conductance and sap-flux-derived transpiration
rates by trees and palms in a subset of tree islands in EFForTS-BEE (Ellsäßer et al., 2020a). However, one central problem
with using UAV-based observations to analyze spatial differences in ET is the variability of weather conditions, such as solar
irradiance and air temperature, during the observation period. This can either be addressed by only analysing ET data recorded
under specific, comparable climatic conditions (e.g., high evaporative demand), or by using the evaporative fraction (EF), i.e.
the ratio of latent heat flux to available energy, or the ratio of actual ET to potential ET (ETa/ETpot), as a target variable. Both
ETa/ETpot and EF are less susceptible to differences in available energy, and as such, within typical ranges, fluctuate less with
varying weather conditions than ET.



The effects of planted tree diversity and tree island area on ET and EF remain largely unexplored. While both variables have
been linked to ET in previous studies (Baldocchi, 2005; Zhang et al., 2022), they are not recognized as direct drivers of ET or
EF in oil palm landscapes, including the EFForTS-BEE experiment. In urban settings as well as in Neotropical natural forests
and plantations, tree diversity was reported to enhance transpiration (González-Espinosa et al., 2004; Kunert et al., 2012; Wang
et al., 2021). For island area, it was found that vapor pressure deficit (VPD) and water stress varied between small urban tree
islands and larger urban forest areas (Cregg and Dix, 2001). The effects may indirectly be driven by other factors, such as the
structure or tree height variability of a given stand, which may affect turbulent energy exchange. Ehbrecht et al. (2017) reported
a significant relation between stand structure and microclimatic variables such as daily VPD amplitude. Tree height is known
to be related to forest thermal balance as well (Barbeta et al., 2023) and may be used as an indicator of surface roughness.
Vegetation health, expressed in indices such as the NDVI, was also shown to affect ET, and its integration into ET modeling
improved the quality of predictions (Yang and Wang, 2011; Er-Raki et al., 2013). Thus, we expect planted tree diversity and
island area to have indirect effects on ET or EF via mediator variables such as observed woody plant diversity, vegetation
stand structural complexity, tree height variability, and vegetation health.

In our study, we tested the effects of tree diversity and island area on ET and EF across 52 tree islands, ten years after their
establishment, using UAV-based thermography and subsequent energy balance modeling.
We hypothesized that EF increases with increasing
H1: planted tree species diversity,
H2: area of tree islands,
H3: observed diversity of woody species,
H4: stand structural complexity,
H5: variability in tree height, and
H6: vegetation health.

For a conceptual structural equation model, relationships, and assumed mechanisms please see the supplementary materials
(Fig. S1, Table S1).

**2 Methods**

**2.1 Study site**

The study was conducted in the lowlands of Sumatra, Indonesia in Jambi province, near the village Bungku. The experiment
EFForTS-BEE (103.25° E, 1.95° S) is situated in an oil palm monoculture owned by PT. Humusindo Makmur Sejati. Rainfall
in the region is 2235 mm yr$^{-1}$ and average annual temperature is 26.7 °C; dominant soil types are Acrisols (Drescher et al.,



2016). The natural vegetation would be tropical rainforest as described in Rembold et al. (2017). After rainforest conversion oil palms were cultivated. EFForTS-BEE was established in 2013 with 52 tree islands within an area of oil palm monocultures of 140 ha, then between 6 and 12 years old (Teuscher et al., 2016).

## 2.1 Experimental design

The treatments in EFForTS-BEE consist of planted tree species diversity (0, 1, 2, 3 6 different tree species) and tree island
area (25 m², 100 m², 400 m², 1600 m²) (Fig. 1). Six native multi-purpose tree species (*Parkia speciosa* Hassk., Fabaceae; *Archidendron jiringa* (Jack) I.G. Nielsen, Fabaceae; *Durio zibethinus* L., Malvaceae; *Peronema canescens* Jack, Lamiaceae; *Rubroshorea leprosula* (Miq.) P.S.Ashton & J.Heck., Dipterocarpaceae; *Dyera polyphylla* (Miq.) Steenis, Apocynaceae) were planted. For each tree island area class, there is one control plot with conventional oil palm management, representing the original monoculture palm pattern with no tree planting. Before tree planting 40% of oil palms were removed inside of the 100
m², 400 m² and 1600 m² plots to increase light availability. In addition to tree planting, the experiment also allowed for natural plant recruitment. The 25 m² plots were placed between palm rows and no oil palms were removed. In addition, there are four control plots where oil palm is managed as usual (Teuscher et al., 2016). The experimental design follows the random partitions approach which allows using stepwise linear regression models to analyze diversity gradients (Bell et al., 2009).

## 2.2 UAV data acquisition

UAV flight missions were carried out on 18 days in the early dry season between 09 June 2023 and 11 July 2023. Flights were conducted between 10:00 and 14:00 local time to collect data under peak radiation conditions and also to minimize shadows. As the battery capacity of the UAV limited flight times to approx. 20 minutes per charge, the area of EFForTS-BEE was divided into 5 routes of roughly equal size. Each route covered between 10 and 12 plots, wherein the sequence of the flight routes was alternated among days to ensure that plots were visited at different times of the day throughout the campaign. Flight
routes were planned using the UgCS flight planning software (SPH Engineering, Lativa). Flight speed was set to 5 m s⁻¹ with an 80% overlap of images. Flight altitude was between 90 – 100 m above the terrain, resulting in a pixel size of c. 2 cm for RGB and c. 10 cm for thermal images.

We used a DJI M300 RTK system (SZ DJI Technology Co., Ltd., China) to carry multiple payloads simultaneously during the
flight campaigns. To record land surface temperature, the UAV was equipped with a Zenmuse H20T camera that contains a radiometric thermal sensor with a spatial resolution of 640 x 512 pixels, a spectral coverage of one long-wave infrared band (7.5 - 13.5 μm), and a sensitivity of ≤ 50 mK at f/1.0 (DJI, 2021) (Fig.1). For measuring shortwave radiation, the UAV carried an Apogee SP-710 albedometer (Apogee Instruments, Inc., USA), consisting of an upward facing (SP-510) and a downward facing (SP-610) thermopile pyranometer. The recorded irradiance values were stored on an Apogee AT-100 microCache
Bluetooth logger. For recording air temperature and relative humidity, the UAV was equipped with an iButton DS1923-F5 Hygrochron sensor (iButtonLink LLC, USA). To acquire multispectral images, the Zenmuse H20T was swapped with a





MicaSense red-edge MX (MicaSense, Inc., USA), which captures five spectral bands (blue [459-491 nm], green [546.5-573.5 nm], red [660-676 nm], red edge [711-723 nm], near infrared [813.5-870.5 nm]) at a spatial resolution of 1280 x 960 pixels and a spectral resolution of 12-bit. It is equipped with an upwards oriented downwelling light sensor (MicaSense DLS 2),

which records global lighting changes to correct reflectance values of each band (MicaSense, 2020). Both cameras were triggered by the overlap mode, which uses the internal GPS to ensure enough overlap of pictures at flight altitude.

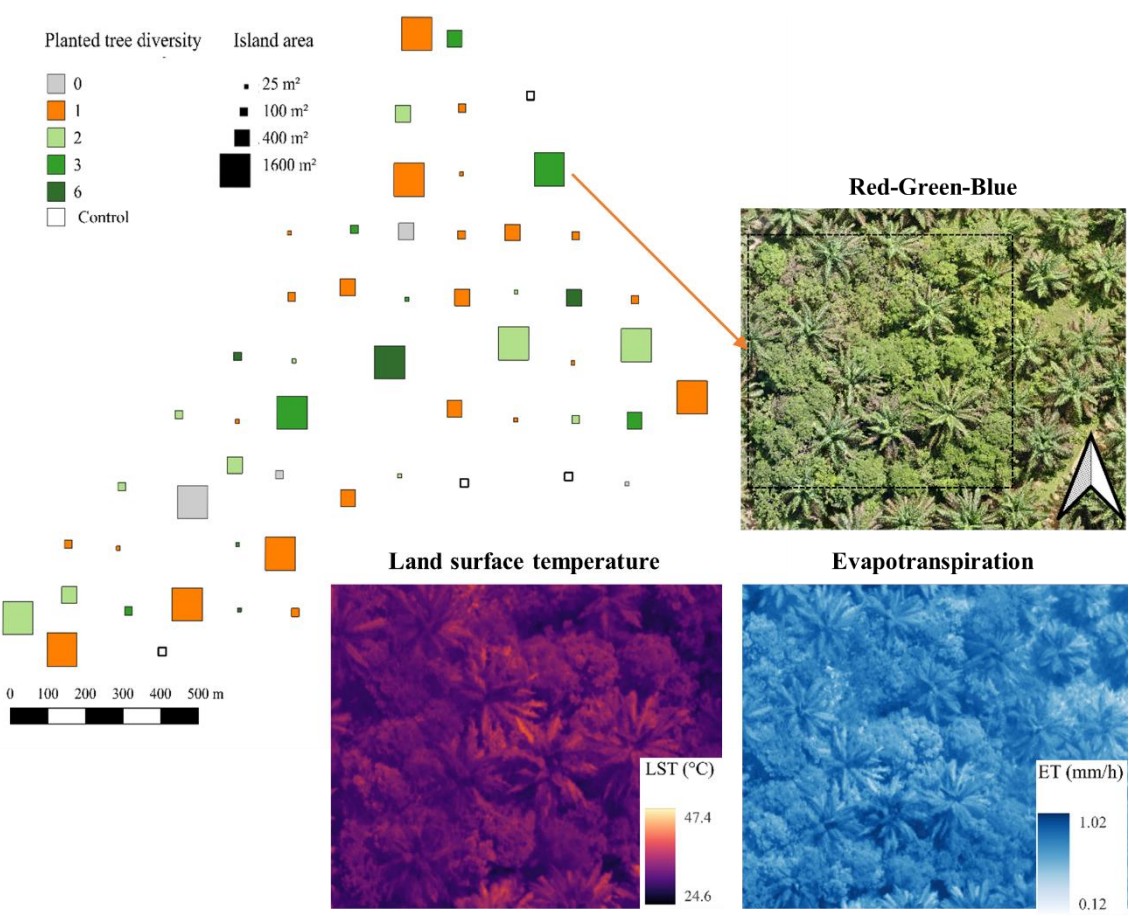


**Figure 1: The experimental design of EFForTS-BEE with 52 tree islands varying in planted tree diversity and island area (island size not drawn to scale). The islands are dispersed in a monocultural oil palm plantation of 140 ha. Red-Green-Blue (RGB), land surface temperature (LST, °C) and evapotranspiration (ET, mm h$^{-1}$) of one large tree island (1600 m$^2$) with three planted tree species.**



### 2.3 UAV-data processing

Raw 3-band JPG image files from the thermal camera were converted to 1-band TIFF format using the dji_h20t_rpeg_to_tif R script by Kattenborn (2023) translating them to land surface temperatures. Subsequently, all images of a given mission were processed to one thermal orthomosaic map displaying land-surface temperatures using the photogrammetry software Agisoft Metashape Professional (Agisoft, Russia). The orthomosaics were then re-sampled to a ground resolution of 11 cm and the individual plots were clipped from the orthomosaics using QGIS version 3.34 (QGIS Development Team 2025). Orthomosaics of the single images from the multispectral camera were processed using Pix4Dmapper (Pix4D S.A., Switzerland), therein applying automated image correction from the DLS 2 radiation sensor and respective calibration images that were captured on the ground before each flight (MicaSense, 2020).

### 2.4 Energy-balance modeling

For the energy-balance modeling, we used the QGIS plugin QWaterModel version 1.5 (Ellsäßer et al., 2020b), which implements the DATTUTDUT model by Timmermans et al. (2015). This model was successfully tested against eddy covariance data in an oil palm plantation in the same region (Ellsäßer et al., 2021) and was subsequently also used for other land-use types in the region (Bulusu et al., 2023; Cortés-Molino et al., 2024). DATTUTDUT is a one source energy balance model that defines the maximum and minimum land surface temperature (LST) in an image based on the $100^{th}$ and $0.5^{th}$ percentiles of pixel values, respectively. This percentile-based approach avoids relying on individual pixel values, thereby reducing the influence of outliers or noise. The assumption is that cold pixels indicate high evapotranspiration while hot pixels indicate very low to no evapotranspiration (Timmermans et al., 2015). Incoming shortwave radiation as measured by the on-board sensors is a further key input for QWaterModel (Brenner et al., 2018; Ellsäßer et al., 2020b; Cortés-Molino et al., 2024). Additionally, mean air temperature during flight was given to QWaterModel, while all other settings remained at default values. As output, QWaterModel generates a 6-band raster file: net radiation Rn (W $m^{-2}$), latent heat flux LE (W $m^{-2}$), sensible heat flux H (W $m^{-2}$), ground heat flux G (W $m^{-2}$), evaporative fraction EF and evapotranspiration ET (mm $h^{-1}$) (Ellsäßer et al., 2020b). Therein, EF is the proportion of available land surface energy that is used for LE. It is calculated as EF = LE / (LE + H). Out of all conducted flights, 43 met the quality criteria, which were clear skies, high incoming radiation and complete, well aligned orthomosaics. The number of observations per tree island ranged from 5 to 10. The observation per tree island with the highest radiation was used for analyzing the controlling factors of EF. 98% of these selected observations (n = 52) were obtained under high incoming shortwave radiation (850 - 1000 W $m^{-2}$). In contrast to ET, EF is only marginally affected by varying levels of radiation (Fig. A1). For both ET and EF, the mean of all pixels within a given tree island were used for the analyzes. By design, EF and ET and their mean values over the area of a given tree island, are independent of scale. To confirm this assumption, tree islands were sub-sampled and the effect of island area within artificially reduced islands was tested (see supplementary materials table S2).



## 2.5 Vegetation health

Multispectral images were used to create orthomosaics of vegetation indices relating to vegetation health (Hernández-Clemente et al., 2019). The following analyzes use the Green Normalized Differential Vegetation Index (GNDVI) since it is sensitive to chlorophyll contents at medium to high LAI (Gitelson and Kaufman, 1998). As described by Gitelson et al. (1996), GNDVI was calculated as:

$$GNDVI = \frac{(NIR - Green)}{(NIR + Green)} \, .$$

## 2.6 Observed woody plant diversity

Data on observed woody plant diversity, including planted as well as naturally regenerated plants, were recorded in 2022. The inventory included all free-standing woody plants with a height greater than 130 cm from the soil surface. Specimens were identified as described in Paterno et al. (2024). A total of 77 species (excluding planted trees) were recorded, with a plot mean of 8.35 (SD 6.37). Hill number q1 was used as a measure of observed woody plant diversity; H1 corresponds to the exponential Shannon entropy also referred to as "typical" diversity (Chao et al., 2014). The R package "hillR" version 0.5.2 (Li, 2018) was used to calculate H1.

## 2.7 Tree height

Maximum tree height and oil palm meristem height were measured for all woody plants in 2022, i.e., oil palm, planted trees and natural regeneration. For heights lower than 2 m, we used a standard tape measure, between 2 and 8 m a measuring rod and for trees taller than 8 m a Vertex (Haglöf, Sweden) device (Zemp et al., 2019a). Average tree height was 5.07 m (SD 3.93) while the oil palms had an average meristem height of 7.8 m high (SD 1.13). The Gini index of tree height was calculated using the R package "ineq" version 0.2-13 (Zeileis, 2014); it is a measure of the unevenness of tree height and serves as a proxy for surface roughness of the stand in our study.

## 2.8 Stand structural complexity

Terrestrial laser scanning (TLS) was carried out in 2022 using a FARO Focus M70 (Faro Technologies Inc., Lake Mary, USA) laser scanner. A stand structural complexity index (SSCI) was calculated as described by Ehbrecht et al. (2017). These data were previously published by Kikuchi et al. (2024). SSCI captures the structural complexity of forest stands by combining the mean fractal dimensions (MeanFrac) with the effective number of layers (ENL) (Ehbrecht et al., 2017). A higher SSCI may reflect a greater surface area available for ET, which could promote higher energy partitioning towards latent heat, therefore SSCI could be linked to EF.



### 2.9 Statistical analyzes

The effects of island area, tree species richness (linear and nonlinear effects) and tree species identity on EF were tested using
a random partitions analysis (Bell et al., 2009; Teuscher et al., 2016). Assumptions for linear models (normal distribution of
residuals) were checked for the response variable EF via the quantile–quantile plots using the package "stats" version 4.3.2
(Team, 2023). The assumption of heteroscedasticity was checked using the "performance" package version 0.13.-0 (Lüdecke
et al., 2021). EF was logit transformed following recommended practices (Warton and Hui, 2011; Li et al., 2023). The overall
random partitions model is stated as:

$$y = \beta_0 + \beta_{LR}X_{LR} + (\sum_i^6 \beta_i x_i) + \beta_{NLR}x_{NLR} + \beta_p x_p + e, \quad (1)$$

where y is the response variable, in our case EF; $\beta_0$ is the model intercept; $X_{LR}$ is planted enrichment tree-species-richness
treated as a continuous variable ("linear richness"); $x_i$ is an indicator of the presence or absence of a certain species (*i*) from
the pool of six planted tree species; $x_{NLR}$ is the "non-linear richness", which is the effect of species richness levels as factors;
$x_p$ is island area as a factor; and $e$ is a normally-distributed error term (Li et al., 2023). In EFForTS-BEE, the random partitions
model focused on 48 plots; the no tree planting plots and the "business-as-usual" islands were not part of the analysis, since
they do not contribute to the species richness gradient. All analyzes were performed in R, using code developed by Li et al.
(2023).

Structural equation modeling (SEM) was used to investigate the direct and indirect effects of the treatments (planted tree
diversity and island area) on EF. Hypotheses to be tested are the direct influences of planted tree diversity and island area on
EF. Further hypotheses include potential indirect effects via endogenous variables. In our study, the endogenous variables
observed woody plant diversity, vegetation stand structural complexity, tree height variability and vegetation health are
considered. Endogenous variables may act as mediators, indirectly linking planted tree diversity, island area, and observed
woody plant diversity to EF. Piecewise SEM are made up of several linear models, which are:
Model 1 = lm(Observed woody plant diversity ~ Planted tree diversity + Island area)

Model 2 = lm(Structural complexity ~ Planted tree diversity + Island area + Observed woody plant diversity)

Model 3 = lm(Tree height variability ~ Planted tree diversity + Island area + Observed woody plant diversity)

Model 4 = lm(Vegetation health ~ Planted tree diversity + Island area)

Model 5 = lm(EF ~ Planted tree diversity + Island area + Observed woody plant diversity + Structural complexity + Tree
height variability + Vegetation health)

The initial models 1, 2, 3 and 4 were built based on prior knowledge and assumptions, following the "weight of evidence"
approach (Grace, 2020). We assumed that structural complexity, observed woody plant diversity, vegetation health and tree
height variability are directly influenced by the treatments, planted diversity and island area. For model 5, we assumed that
differences in evaporative fraction are driven by the treatments as well as by all endogenous variables. The endogenous
variables were not correlated among each other (spearman correlation < 0.5, see Fig. A2). The assumption of normality of



residuals was tested using the Shapiro–Wilk test. Normality could not be assumed, so EF was transformed (standardized Box Cox transformation) using the "bestNormalize" R package version 1.9.1 (Peterson, 2021). For the calculation of standardized effect sizes in the structural equation model, all variables except – planted tree diversity and tree island area – were standardized using the "vegan" package version 2.6-4 (Dixon, 2003). The SEM was built using the "pievewiseSEM" package version 2.3.0

(Lefcheck, 2016). Non-parametric bootstrapping with 1000 randomizations was used to calculate effects including confidence intervals for each predictor, using the "semEff" package version 0.6.1 (Murphy, 2022). The "business as usual" plots are not included in the SEM, because their continued management might introduce confounding effects and we decided to focus on tree islands only. Therefore, the SEM focused on 52 plots, out of the initial 56.

## 3. Results

The mean ET across the 52 islands was 0.74 mm h$^{-1}$ (SD: 0.09 mm h$^{-1}$) at the time the thermal images were recorded. The minimum ET observed was 0.49 mm h$^{-1}$, and the maximum was 0.95 mm h$^{-1}$. The mean EF was 0.7 (SD: 0.06), with values ranging from 0.56 to 0.83. GNDVI ranged between 0.57 and 0.86.

### 3.1 Random partitions model: effects of planted tree diversity and tree island area

No significant effects of continuous richness, species identity, or non-linear richness on EF were detected using the random

partitions model. However, tree island area had a significant positive effect ($p < 0.05$) on EF (Figure 2, Table 1).

**Table 1: Controls of evaporative fraction (EF) as indicated by a random partitions model. EF was Logit transformed and unit variance scaled.**

| Effect | Df | Sum Sq | Mean Sq | F value | Pr (>F) |
|---|---|---|---|---|---|
| Richness (continuous) | 1 | 0.0055 | 0.0055 | 0.0104 | 0.9192 |
| Species identity | 5 | 4.1545 | 0.8309 | 1.5907 | 0.1876 |
| Richness (categorical) | 2 | 1.1564 | 0.5782 | 1.1069 | 0.3416 |
| Island area | 3 | 22.879 | 7.6263 | 14.6 | < 0.05 |
| Residuals | 36 | 18.8046 | 0.5224 | | |

Specifically, significant effects ($p < 0.05$) were observed for island areas of 25 m$^2$, 100 m$^2$ and 1600 m$^2$. With each increase in tree island area, the standardized effect on EF rose by approximately 0.5. From the smallest island area (25 m²) to the largest (1600 m²), EF increased by 17%, from an average of 0.65 to 0.76.




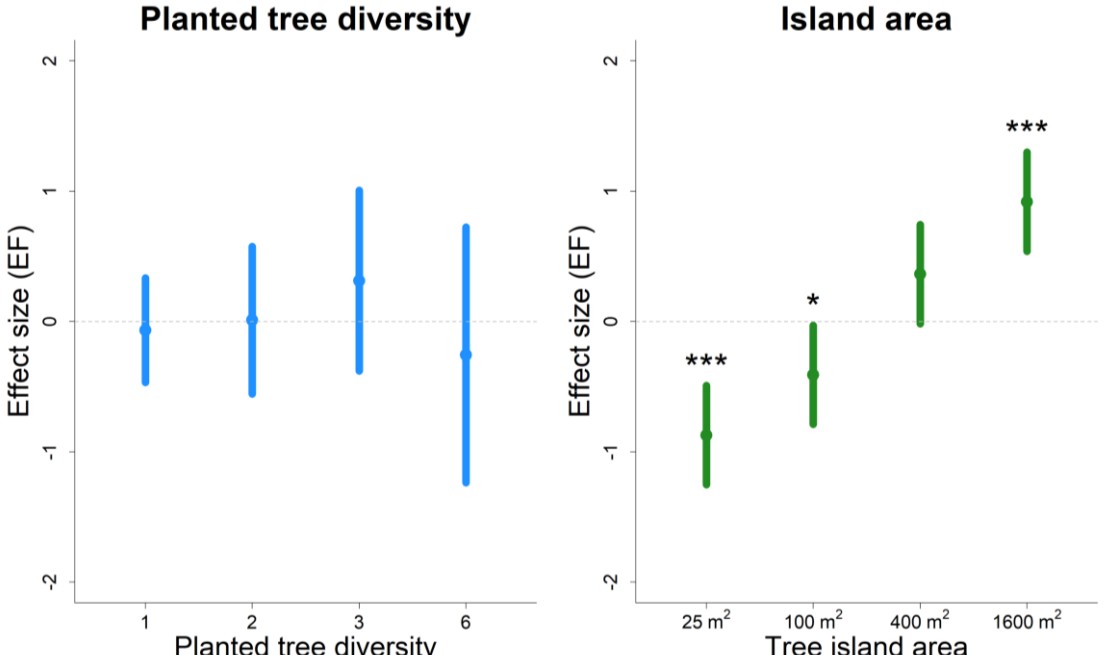

**Figure 2: Standardized effect sizes of planted tree species diversity and tree island area on evaporative fraction as derived from the random partitions model. Effect sizes are shown with 95% confidence intervals, N = 48. Asterisks indicate p-value levels: *** p < 0.001, ** p < 0.01, * p < 0.05. Coefficients and p-values are based on a sequential linear model (Bell et al. 2009).**

### 3.2 Structural equation model: direct and indirect effects

Planted tree diversity had no significant direct or indirect effects on EF, challenging hypothesis H1. In contrast, hypothesis H2 was supported, as tree island area had a strong positive direct effect on EF (Std.Beta = 0.44, CI[0.28-0.64], p < 0.001). Island area also exhibited an indirect positive net effect on EF. Specifically, island area strongly and positively affected observed woody plant diversity (Std.Beta = 0.8, CI[0.69-0.87], p < 0.001), which in turn had a positive effect on structural complexity (Std.Beta = 0.28, CI[0.04-0.51], p < 0.05), ultimately enhancing EF (Std.Beta = 0.2, CI[0.06-0.38], p < 0.05). This path of positive effects was partly offset by a negative direct effect of island area on structural complexity (Std.Beta = -0.33, CI[-0.55 - -0.06], p < 0.05), though this effect was much smaller in magnitude (Figure 4). Hypothesis H3 and H4 – which predicted positive effects of observed diversity of woody species and stand structural complexity on EF – were both supported by significant SEM paths. Observed woody plant diversity positively influenced stand structural complexity, which in turn positively affected EF. Contrary to hypothesis H5, tree height variability had no significant effect on EF. Similarly, hypothesis H6 was not supported, as vegetation health showed no significant relationship with EF. Additional results not predicted by any hypothesis included a significant direct path from planted tree diversity to tree height variability (Std.Beta = 0.36, CI[0.14-0.52], p < 0.001) and a significant direct path from observed woody plant diversity to tree height variability (Std.Beta = 0.25, CI[0.06-0.5], p < 0.05).




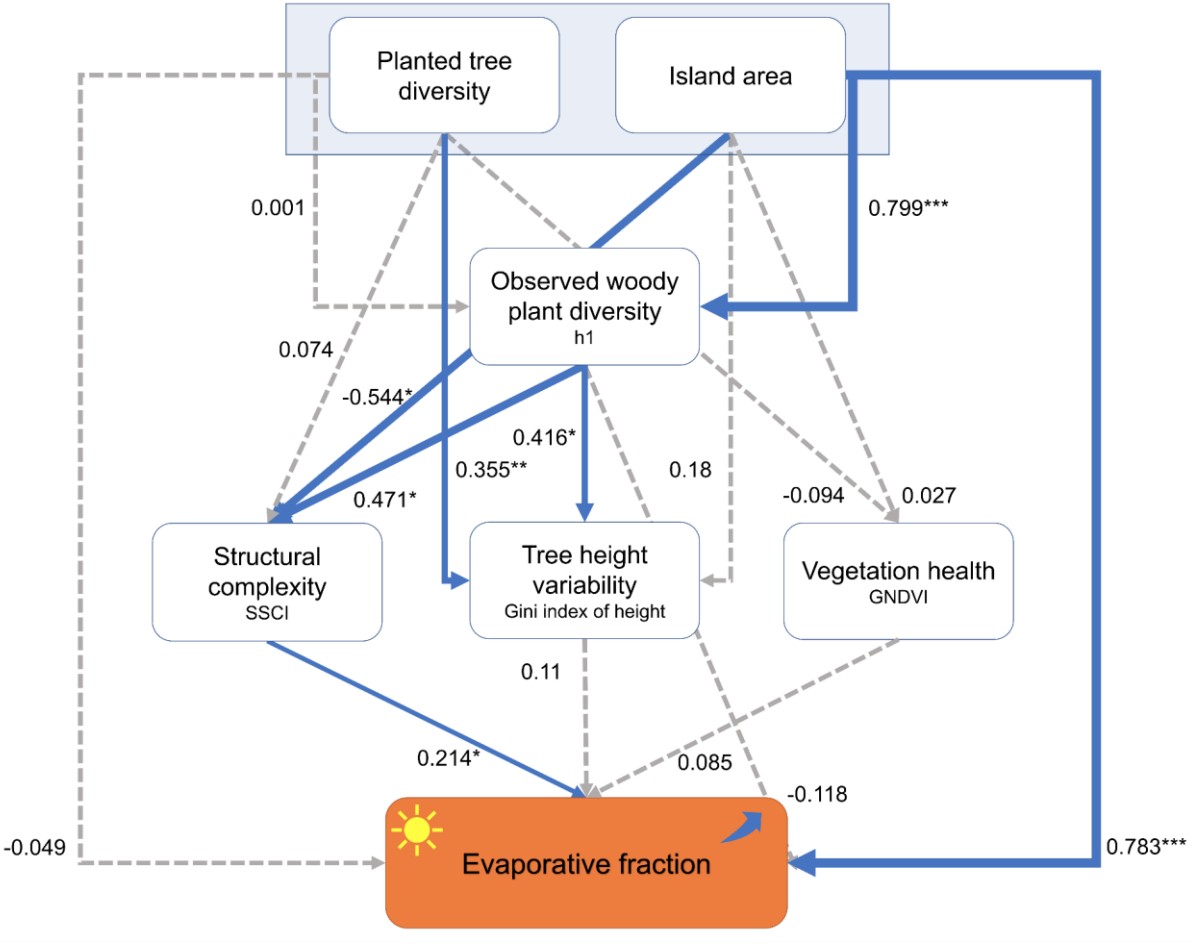

350 **Figure 3: Drivers of evaporative fraction as indicated by a piecewise structural equation model (N = 52, Fisher's C = 2.01, d.f. = 8, P = 0.98). Solid lines indicate significant path coefficients (p < 0.05). Symbols indicate p-value levels: * p < 0.05 **, p < 0.01 ***, p < 0.001.**

355





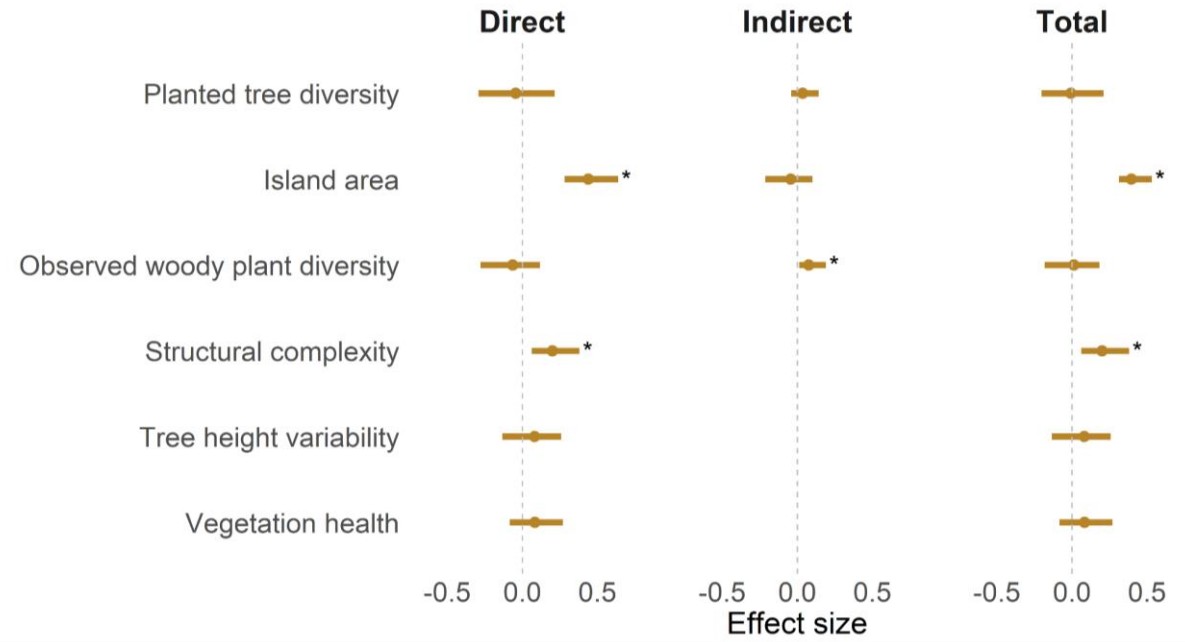

**Figure 4: Standardized direct, indirect and total effect sizes of predictors on evaporative fraction in SEM. Error bars show 95% CI, derived through bootstrapping with 1000 iterations. Symbols indicate p-value level: * p < 0.05.**

## 4. Discussion

Our UAV-based assessment of 52 tree islands in an oil palm landscape revealed that tree island area had a direct positive effect on EF. Additionally, tree island area had an indirect positive effect on EF via observed woody plant diversity and structural complexity. Whilst planted tree diversity did not have any direct or indirect effects on EF, observed woody plant diversity had an indirect positive effect on EF via structural complexity.

## 4.1 ET and EF in context

The mean ET during high incoming radiation conditions from the tree islands in EFForTS-BEE was 0.74 mm h$^{-1}$, with an evaporative fraction of 0.70. Data were filtered for high incoming radiation conditions (> 850.00 W m$^{-2}$) to ensure comparability among islands. These ET values are similar to those reported in other UAV-based studies in the same region. For example, in a mature and intensively managed oil palm plantation an ET of 0.78 mm h$^{-1}$ was observed around noon, and the UAV-based method performed well compared to the eddy covariance method (Ellsäßer et al., 2021). Additional UAV-derived data from rain forest sites in the region showed ET rates between 0.5- and 0.8-mm h$^{-1}$ under favorable atmospheric conditions, with plot-level variation attributed to site conditions (Bulusu et al., 2023). Overall, the observed ET rates and EF from the tree islands fall within the regional range and may be at the higher end of the spectrum compared to other land-use




types. This is consistent with other studies indicating high evaporation or transpiration rates from early secondary forests and plantations in the tropics (Hölscher et al., 1997; van Dijk and Keenan, 2007; Dierick et al., 2010).

**4.2 Direct and indirect effects of planted tree diversity and island area (H1 & H2)**

In EFForTS-BEE, we did not find significant effects of planted tree diversity on ET or EF. This contrasts with widespread observations of increasing ecosystem functions with greater tree diversity, as reported for aboveground productivity (Zheng et al., 2024). Data on ET are scarce, but results from the Sardinilla tree diversity experiment in Panama indicate an increase in transpiration with tree diversity (Kunert et al., 2012). However, planted tree diversity does not always lead to higher EF. For EFForTS-BEE, Kikuchi et al. (2024) reported a high mortality rate (> 90%) for *D. polyphylla*, *D. zibethinus* and *R. leprosula* and a moderately high mortality rate (60%) for *P. speciosa* 9 years after planting. These mortality rates in four of the six planted species lead to lower realized planted tree diversity, possibly weakening the effects of planted tree diversity. Additionally, the presence of oil palms in all plots, abundant natural recruitment of woody species, and island area may have diluted the effect of planted tree diversity. Notably, planted tree diversity did show a significant positive effect on tree height variability, consistent with the tree Height Variation Hypothesis (VHV) (Torresani et al., 2023), which uses tree height heterogeneity as a remote sensing-based biodiversity indicator. Marks et al. (2016) also reported a strong positive correlation between tree species richness and maximum tree heights in forests of eastern as well as western North America.

Island area was a strong predictor of EF, as indicated by both the random partitions model (Fig. 2) and SEM. EF increased significantly with increasing island area from 25 m$^2$ to 1600 m$^2$ by 17%. In the SEM (Fig. 3), tree island area had both direct and indirect effects on EF. The direct path was associated with at high standardized effect size (Std.Beta = 0.44, p < 0.001) (Fig. 4). This tree-island area effect may be attributed to higher EF at the edge between the tree island and the surrounding monoculture oil palm plantation. Significant edge effects on EF were detected in the 400 m$^2$ and 1600 m$^2$ tree islands, with EF decreasing along the distance-to-edge gradient (see supplementary materials table S3). This pattern aligns with findings by Giambelluca et al. (2003), showing an increased transpiration at the edge of a secondary forest in Vietnam. Using similar methods in the Brazilian Amazon, Kunert et al. (2015) likewise found enhanced transpiration at edges. However, other studies have reported the opposite. In a model-based study in the southwestern Amazon, Numata et al. (2021) found no ET difference between forest edges and interiors under high water availability. As dry seasons progressed, ET declined more near edges (up to 100 m) than in interior forests, especially during extreme droughts. Another modeling study in the Brazilian Atlantic Forest region reported that ET was reduced by 43% at the edges of forest fragments when compared to the core (Dantas de Paula et al., 2015). We conclude that edges are expected to induce changes in ET or EF, although the direction of change in other studies was variable. The variability could among others be induced by the origin of the forest patch as a remnant of former old-growth forest with a biomass decay at the edge (Ordway and Asner, 2020) or recovering secondary forest, the adjacent vegetation, and energy advection e.g., by wind. In our experiment, conditions were similar among tree islands, supporting the observed increase in ET and EF with tree island area.





On the indirect path, tree island area showed a strong positive path coefficient ($R^2 = 0.8$, $p < 0.001$) for observed woody plant diversity. This result aligns with a previous study in the same experiment (Paterno et al., 2024), which showed that increasing tree island area leads to higher taxonomic diversity of recruiting woody species. This result reflects the species-area relationship (SAR), were larger areas support greater diversity (Drakare et al., 2006). However, when accounting for sampling effort, woody plant diversity increases with tree island area, probably due to greater environmental heterogeneity in larger islands that supports a broader range of species and ecological strategies (Paterno et al., 2024). The SEM pathway continues from observed woody plant diversity to structural complexity. Earlier, 3 years after planting, Zemp et al. (2019b) found a non-linear increase of SSCI with planted tree diversity. Later, 9 years after planting, Kikuchi et al. (2024) found that diversity and size variation of recruiting trees became more important, as planted tree diversity had no effect on SSCI. Applying a structural equation model, Ma et al. (2024) found a significant positive indirect path linking tree species richness to coarse wood productivity in forests across a latitudinal gradient in China. Mean height, used as an indicator of vertical structure in their study, served as the mediator in this indirect path. In young temperate forest plots, a significant positive indirect effect of tree species richness on annual wood productivity (AWP) was identified (Ray et al., 2023). As in this study, SSCI acted as a mediator in their SEM. In a study in the Sardinilla tropical tree diversity experiment, Ray et al. (2024) found a positive direct significant path from tree species richness to canopy space filling (CSFI). In their SEM, CSFI was linked by a significant positive path to wood productivity, showing an indirect positive effect of tree species richness on wood productivity. Their result is in line with the outcome of our study, as both report an indirect path from diversity via structure towards an indicator of productivity. It is however acknowledged that the patterns and mechanisms deserve further scientific attention.

**4.3 Direct effects of observed diversity, vegetation structure and vegetation health (H3 – H6)**

Direct and indirect effects of the treatments planted tree diversity and island area on EF were tested by structural equation modeling. Direct effects of observed woody plant diversity, stand structural complexity derived from laser scanning, tree height variability and plant health (as indicated by GNDVI) on EF were also tested.

Ten years after establishment of EFForTS-BEE, we did not find a direct effect of observed woody plant diversity (H3) (Fig. 3) on energy partitioning. Across EFForTS-BEE, a total of 84 woody species were observed, consisting of the six planted tree species complemented by 78 woody species originating from natural recruitment. Despite one plot showing no species (Shannon Index = 0), the interquartile range (0.99 - 1.87) and a median of 1.45 suggest a relatively uniform distribution of moderate to high woody plant diversity across tree islands, reflecting the presence of a wide range of species throughout EFForTS-BEE. Our finding that observed woody plant diversity enhances EF indirectly through stand structural complexity aligns with diversity - ecosystem function (BEF) theory (Morin et al., 2011), supporting the idea that diversity-driven complementarity – through more efficient 3D - space filling (Jucker et al., 2015; Juchheim et al., 2017) – can strengthen ecosystem functioning even at moderate to high diversity levels.

A significant positive effect of structural complexity on EF was detected ten years after tree island establishment (Fig. 3). When modeling ET in boreal ecosystems, Leonard et al. (2022) integrated forest stand complexity variables into their model





and thereby reduced unexplained variance by 10%. Running their Boreal Ecohydrological Tree Algorithm (BETA+) model (Leonard et al., 2021) with variables describing the stand structure, their ET estimate increased by 26%. This is consistent with the positive effect size of SSCI on EF in our SEM. Enhanced stand structure may change the microclimate and increase energy exchange.

However, in EFForTS-BEE we did not find a significant effect of the Gini index of tree height on energy partitioning toward

latent heat flux. Canopy height is an axis of surface roughness in forests (Maurer et al., 2015). In a study on the biogeophysical effects of deforestation, a decrease of surface roughness led to decreased evapotranspiration in the tropics (Winckler et al., 2019). In eucalypt forests of south-eastern Australia, Meili et al. (2024) attributed fast ET recovery after fire to aerodynamic warming from shorter forest height, highlighting the importance of height in ecohydrology. In urban landscapes, vertical canopy structure was a key predictor of LST (Chen et al., 2020). Tree height variability may not serve as a substitute for surface

roughness in the tree islands of EFForTS-BEE, which could explain the lack of a significant effect on EF. As some of the planted trees, as well as the natural regeneration, is below oil palm canopies, tree height variability might not be the dominating factor of surface roughness.

Despite our assumption, vegetation health (GNDVI) did not predict EF. Vegetation indices are often used as calibration factors in ET modeling. For example, the space borne level-3 evapotranspiration (ET_PT-JPL) product uses normalized difference

vegetation index (NDVI) and soil adjusted vegetation index (SAVI) to convert potential ET to actual ET (Fisher, 2018). Yang and Wang (2011) successfully modeled EF based on spaceborne LST maps using a combined approach of the Priestly-Tylor equation, NDVI and day-night LST differences. Both studies however are addressing larger spatial scales with space borne images, on such larger scales stronger differences in vegetation indices between different land-use types are to be expected. In EFForTS-BEE, the tree islands comprise of dense vegetation at a relatively good health status across all tree islands.

**5. Conclusion**

Tree-island area enhances evaporative cooling through both direct and indirect pathways, primarily by increasing woody plant diversity and structural complexity. A decrease in EF along the edge gradient found in larger tree islands may explain this tree-island-area effect. Other factors – planted tree diversity, tree height variability, and vegetation health – did not show significant effects on EF, possibly because natural regeneration has become more influential than the initially planted trees at this stage

of the experiment. These findings indicate that tree-island area, observed woody plant diversity, and structural complexity are important factors enhancing energy partitioning toward latent heat flux, thereby cooling the environment. Further research is needed to better understand the mechanisms behind the tree-island-area effect on evaporative fraction. Our results suggest that larger and structurally more complex tree islands play a role in regulating microclimate in human-modified landscapes. These findings provide a basis for integrating tree islands into agricultural landscapes to improve evaporative cooling and climate

regulation.

## Acknowledgments

We thank PT. Humusindo for allowing us to work on their land. We further thank our research assistant Tiarma Rezeki Manalu for her support during data collection. The study was founded by the DFG (Deutsche Forschungsgemeinschaft, German Research Foundation), in the framework of the collaborative German-Indonesian research projects CRC990 EFForTS (project
ID 192626868), ClimReg (project ID 532868192), and EFForTS-BEE (project ID 532776526). This study was conducted using research permits: no. 212/SIP/IV/FR/10/2022 (Thorge Wintz) and no. 100/SIP/IV/FR/2/2023 (Gustavo Brant Paterno). ChatGPT was used to generate parts of the R code used for data analysis.

## Code / Data availability

The final data set including ET and EF values used for statistical tests were uploaded to Göttingen Research Online database
(https://doi.org/10.25625/Q7B9TR). Raw thermal and multispectral orthomosaics, QWater raster files and R code used for modeling are available upon request to the corresponding author.

## CRediT author contributions

**Thorge Wintz:** Conceptualization, Investigation, Formal analysis, writing – original draft, writing – review & Editing. **Alexander Röll:** Conceptualization, Writing – Review & Editing, Supervision. **Gustavo Brant Paterno:** Methodology,
Formal analysis, Writing – Review & Editing. **Florian Ellsäßer:** Writing – Review & Editing. **Delphine Clara Zemp:** Methodology, Formal analysis, Writing – Review & Editing. **Hendrayanto:** Project administration, Writing – Review & Editing. **Bambang Irawan:** Project administration, Writing – Review & Editing. **Alexander Knohl:** Writing – Review & Editing, Supervision. **Holger Kreft:** Writing – Review & Editing, Supervision. **Dirk Hölscher**: Conceptualization, Writing – Review & Editing, Supervision, Project administration, Funding acquisition.

## Competing interests

There are no conflicts of interest, financial or otherwise.



**Appendix**


For a conceptual structural equation model, relationships and assumed mechanisms please see the supplementary materials (Fig. S1, Table S1).

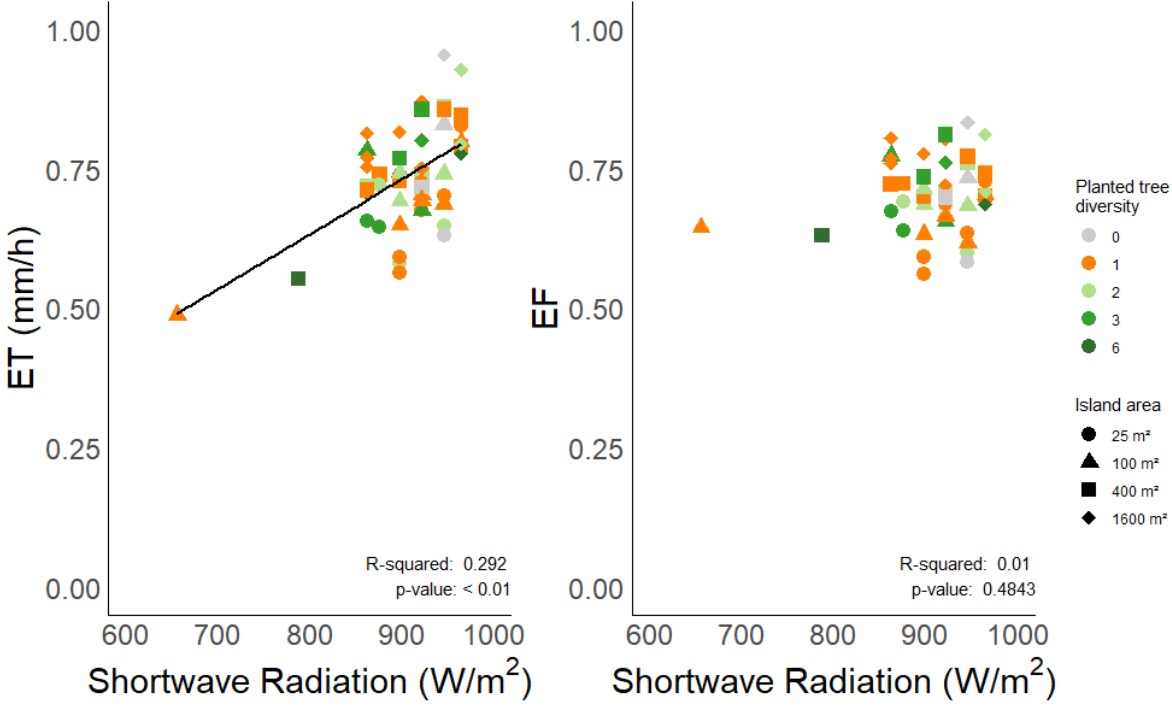


**Figure A1: Evapotranspiration (ET) and evaporative fraction (EF) for the selected observations (one record per island with the highest radiation during the flight; n = 52) against shortwave incoming radiation during the flight. Planted tree diversity is indicated by different symbols and island is given in different colours. 98% of the observations were obtained at incoming shortwave radiation > 850 W m$^{-2}$. ET increased with incoming shortwave radiation ($R^2$ = 0.29, p < 0.01) whereas EF did not change ($R^2$ 0.01, p = 0.49).**






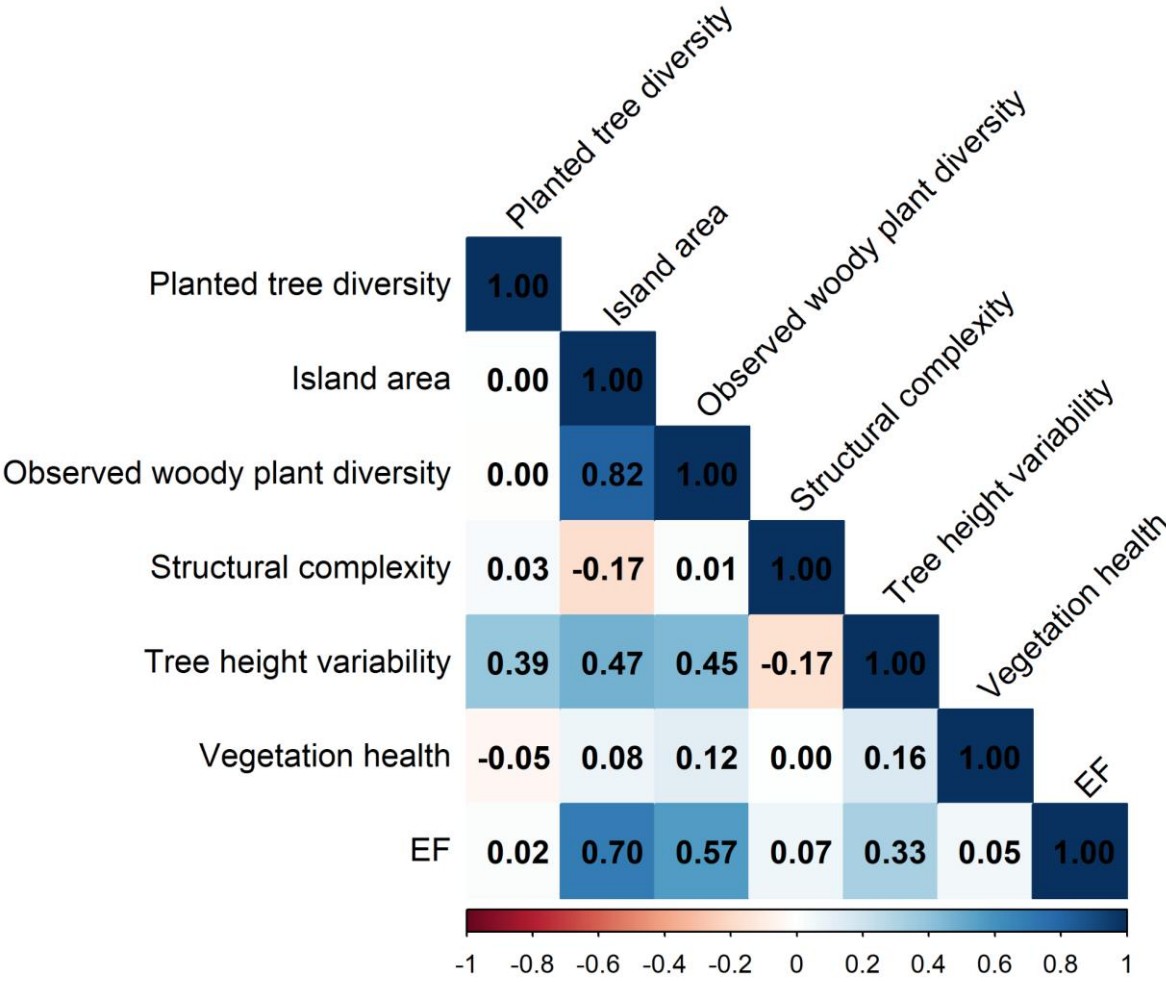

**Figure A2: Correlation plot for variables used in the structural equation model.**






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
