# Peer review of "Tree island area in oil palm agroforests directly and indirectly drives evaporative fraction"

_EGUsphere, 2025_

## Author Comment (AC1)

Thank you for your positive evaluation of our manuscript and for the constructive comments. We believe that your suggestions will help to improve the quality of the manuscript. Specific comments are addressed below.

Specific comment:

First paragraph of section 4.2: This entire paragraph discusses the (lack of) effect of planted tree diversity on EF. I think this discussion is somewhat not relevant and maybe misleading, since the observed tree diversity (the actual tree diversity of the environments in the time of measuring) had a positive effect on EF. The lack of effect of planted tree diversity is just and artifact of the lack of correlation between planted diversity and current diversity (due to mortality and other ecological mechanism not addressed in the manuscript). For example the second sentence: "This contrasts with widespread observations of increasing ecosystem functions with greater tree diversity, as reported for aboveground productivity (Zheng et al., 2024).", is problematic because the authors are talking about planted tree diversity and not actual observed diversity and, additionally, the authors did found indirect effect of tree diversity on EF via structural complexity. My suggestion is to re-write this paragraph focusing on why planted tree diversity did not affected observed tree diversity and consequently EF.

The discussion presented in lines 434-437 contradicts the discussion presented in the first paragraph of section 4.2 (commentary above).

We restructured section 4.2 accordingly to discuss more explicitly the effects of planted and observed tree diversity, this resolves former apparent contradictions.

Revised version of section 4.2:

In EFForTS-BEE, we did not find significant effects of planted tree diversity on ET or EF. It is likely that any influence of planted tree diversity on EF was superimposed by effects of observed woody plant diversity. Paterno et al. (2024) reported a positive effect (P = 0.049) of planted tree diversity on taxonomic diversity of recruiting species six years after establishment; however tree island area showed a much stronger positive effect. The effects of observed woody plant diversity are discussed in the following section (4.3).

Technical corrections:

Standardize the supplementary material naming. You call it supplementary material in the main text, and Appendix at the end. Same for supplementary figure naming (ie.: S1 vs. A1)

Figure S1/A1 is not a "conceptual structural equation model, relationships and assumed mechanisms", ratter a scatter plot ET/EF vs. Shortwave radiation. The authors should include the graphical representation of the conceptual structural equation model, and scatter plots of individual relationships (ie.: diversity vs. ET).

Table S1 is not shown in the appendix.

Figure 3: the path from "Island Area" to "Structural Complexity" can be confused with a path from "Observed woody plant diversity" to "Structural Complexity". Maybe change the figure a bit to make this path more clear.

Line 395. Supplementary material S3 not shown.

We standardized the naming of appendix and supplementary materials according to Biogeosciences guidelines. To improve clarity, we added a reference in the main text to link to the separate Supplementary PDF file, where Figures S1-S3 and Table S1 are provided.

Figure 3 has been slightly modified by interchanging "Vegetation health" and "Structural complexity". This way the path from "Island area" to "Structural complexity" is no longer covered by "Observed woody plant diversity".

[Figure]

*Figure 3: Drivers of evaporative fraction as indicated by a piecewise structural equation model (N = 52, Fisher's C = 2.01, d.f. = 8, P = 0.98). Solid lines indicate significant path coefficients (p < 0.05). Symbols indicate p-value levels: \* p < 0.05 \*\*, p < 0.01 \*\*\*, p < 0.001.*

Scatter plots of the relationships between variables used in the SEM and EF were added to the appendix.

[Figure]

*Figure A3: Relationship between evaporative fraction (EF) and predictor variables. Panels show scatter plots and linear regression for (a) planted tree diversity, (b) island area, (c) observed woody plant diversity, (d) structural complexity, (e) tree height variability, and (f) vegetation health (n = 52). Regression lines indicate significant relationships (p < 0.05).*

Significant relationships between evaporative fraction and predictor variables were added to section 3.

Revised version of section 3:

The mean ET across the 52 islands was 0.74 mm $h^{-1}$ (SD: 0.09 mm $h^{-1}$) at the time the thermal images were recorded. The minimum ET observed was 0.49 mm $h^{-1}$, and the maximum was 0.95 mm $h^{-1}$. The mean EF was 0.7 (SD: 0.06), with values ranging from 0.56 to 0.83. GNDVI ranged between 0.57 and 0.86. Significant linear relationships were found between EF and tree island area (p < 0.001), observed woody plant diversity (p < 0.001), and tree height variability (p < 0.01) (Fig. A3).